# Luminescent Studies on Germanate Glasses Doped with Europium Ions for Photonic Applications

**DOI:** 10.3390/ma13122817

**Published:** 2020-06-23

**Authors:** Jacek Żmojda, Marcin Kochanowicz, Piotr Miluski, Piotr Golonko, Agata Baranowska, Tomasz Ragiń, Jan Dorosz, Marta Kuwik, Wojciech Pisarski, Joanna Pisarska, Renata Szal, Gabriela Mach, Bartosz Starzyk, Magdalena Leśniak, Maciej Sitarz, Dominik Dorosz

**Affiliations:** 1Faculty of Electrical Engineering, Bialystok University of Technology, 45D Wiejska Street, 15-351 Bialystok, Poland; m.kochanowicz@pb.edu.pl (M.K.); p.miluski@pb.edu.pl (P.M.); piotr@neoweb.pl (P.G.); doroszjan@pb.edu.pl (J.D.); 2Faculty of Mechanical Engineering, Bialystok University of Technology, 45C Wiejska Street, 15-351 Bialystok, Poland; a.baranowska@pb.edu.pl (A.B.); t.ragin@pb.edu.pl (T.R.); 3Institute of Chemistry, University of Silesia, 9 Szkolna Street, 40-007 Katowice, Poland; marta.kuwik88@gmail.com (M.K.); wojciech.pisarski@us.edu.pl (W.P.); joanna.pisarska@us.edu.pl (J.P.); 4Faculty of Materials Science and Ceramics, AGH University of Science and Technology, 30 Mickiewicza Av., 30-059 Krakow, Poland; renata.szal@agh.edu.pl (R.S.); machgabriela1@gmail.com (G.M.); bar.s@wp.pl (B.S.); mlesniak@agh.edu.pl (M.L.); msitarz@agh.edu.pl (M.S.); ddorosz@agh.edu.pl (D.D.)

**Keywords:** germanate glass, europium oxide, luminescent properties, spectroscopic probe

## Abstract

Glass and ceramic materials doped with rare earth (RE) ions have gained wide interest in photonics as active materials for lasers, optical amplifiers, and luminescent sensors. The emission properties of RE-doped glasses depend on their chemical composition, but they can also be tailored by modifying the surrounding active ions. Typically, this is achieved through heat treatment (including continuous-wave and pulsed lasers) after establishing the ordering mechanisms in the particular glass–RE system. Within the known systems, silicate glasses predominate, while much less work relates to materials with lower energy phonons, which allow more efficient radiation sources to be constructed for photonic applications. In the present work, the luminescent and structural properties of germanate glasses modified with phosphate oxide doped with Eu^3+^ ions were investigated. Europium dopant was used as a “spectroscopic probe” in order to analyze the luminescence spectra, which characterizes the changes in the local site symmetries of Eu^3+^ ions. Based on the spectroscopic results, a strong influence of P_2_O_5_ content was observed on the excitation and luminescence spectra. The luminescence study of the most intense ^5^D_0_→^7^F_2_ (electric dipole) transition revealed that the increase in the P_2_O_5_ content leads to the linewidth reduction (from 15 nm to 10 nm) and the blue shift (~2 nm) of the emission peak. According to the crystal field theory, the introduction of P_2_O_5_ into the glass structure changes the splitting number of sublevels of the ^5^D_0_→^7^F_1_ (magnetic dipole) transition, confirming the higher polymerization of fabricated glass. The slightly different local environment of Eu^3+^ centers the results in a number of sites and causes inhomogeneous broadening of spectral lines. It was found that the local asymmetry ratio estimated by the relation of (^5^D_0_→^7^F_2_)/(^5^D_0_→^7^F_1_) transitions also confirms greater changes in local symmetry around Eu^3+^ ions. Our results indicate that modification of germanate glass by P_2_O_5_ allows control of their structural properties in order to functionalize the emissions for application as luminescent light sources and sensors.

## 1. Introduction

Many photonic applications are based on active waveguide systems. Currently, active hybrid structures are proposed, in which crystalline phases are introduced within the amorphous matrix using various methods, including thermal treatment [1], direct doping [2], chemical vapor deposition (CVD) [3], and 3D printing [4]. All of these methods require glasses with modified structural properties, ensuring the conditions for obtaining new luminescent properties (from embedded active structures) while maintaining the stability of the hybrid system [5,6,7].

This opens a new direction to develop special glasses and optical fibers with unique structural lattice geometries. enabling effective light control. Recently, many novel photonic materials have combined two glass-forming elements with relatively different phonon energies: antimony–borate [8,9], lead–germanate [10,11], lead–phosphate [12,13], germanate–bismuth [14,15], or antimony–phosphate [16,17]. It is evident that this approach allows transparent and thermally stable glass to be achieved, which is useful in optical fiber fabrication. Additionally, due to the low phonon energy, higher efficiency of radiative transitions in lanthanide ions is achieved. Among various glass-forming oxides, the germanate glasses have been extensively investigated due to their unique material and optical properties, such as their low phonon energy (approximately 800 cm^−1^), good thermal stability, wide transparency window (from 0.35 µm to 5 µm), and good lanthanide ion solubility [18,19,20]. In the 1990s, one of the popular glass systems—GeO_2_-Ga_2_O_3_-BaO (GGB)—was developed as a candidate for optical fibers doped with rare earth ions [21]. Until now, these glasses have been modified by various oxides [22] and fluorides [23], however novel luminescent properties can be proposed. In our earlier works, we showed that the substitution of BaO by Sb_2_O_3_ and TeO_2_ in GGB glasses doped with Eu^3+^ ions led to the polymerization of the structure and ordering processes, along with increasing amounts of modifiers [24,25]. The choice of europium as a well-known “structural probe” allowed investigation of the crystal field symmetry in photonic materials [26]. Additionally, Eu^3+^ is one of the most important lanthanide ions, characterized by strong visible emissions, making it applicable in lighting, biochemical and biomedical sensing, as well as spectral imaging [27,28,29]. The other reasons for the popularity of europium ions in the field of photonics are their relevant emission properties resulting from 4*f*→4*f* transitions, whose positions are practically independent of ligands [30]. Therefore, the Eu^3+^ ions play an important role in the luminescence field, and their strong and narrow orange-red emission have been widely used in phosphors and display devices [31,32,33]. Moreover, due to its relatively simple energy level structure, the efficient excitation of europium is achieved by radiation from the UV spectral range (300–400 nm). Thus, incorporation of europium to different optical materials provides an opportunity to use it as UV-sensitive marker in fingerprint technique, fluorescence diagnosis or biomarkers [34,35,36].

In our current research, we focused on the role of P_2_O_5_ in luminescent and structural properties of GGB glasses doped with Eu^3+^ ions. The phosphorous oxide is characterized by high phonon energy (1200 cm^−1^) and reduces clustering effects via non-bridging oxygen formation, better incorporating rare earth ions promoting more efficient emissions than pure germanate glasses [37]. Even more interesting is that the P_2_O_5_ can be used as a nucleating agent for transparent glass–ceramic materials [38,39]. In view of the above, it is important to find a suitable composition of germanate glass that might be potentially used to produce the glass–ceramic material with functionalized luminescent properties. Taking into account the possible application of the investigated glass in light-emitting devices, the chemical composition of glass was modified by gradually replacing the BaO by P_2_O_5_ and by fixing the Eu^3+^ doping level (0.2 mol.%). Deconvolution of mid-infrared (MIR) absorbance spectra was applied to analyze the strong impacts of P-O^−^ bond vibration and non-bridging oxygen (NBO) on the maximization of the phonon energy of the glass matrix. Luminescence spectra and emission kinetics in the visible range were investigated in two channels of excitation at the 394 (^7^F_0_→^5^L_6_) and 464 nm (^7^F_0_→^5^D_2_) wavelengths. To the possible relation between the local structure of the fabricated germanate glass with the luminescent properties of Eu^3+^ ions, the effect of phosphorous oxide on the profiles of luminescence spectra was analyzed in detail.

## 2. Material and Methods

Glasses with molar composition of 50GeO_2_-10Ga_2_O_3_-(40−x)BaO-xP_2_O_5_-0.2Eu_2_O_3_ (labelled as GGBxPEu), where x = 0, 10, 20, 30, 40 mol.%, were melted in an alumina crucible at 1550 °C for 2 h. The glass melt was poured into a brass plate at room temperature (RT) and then annealed at close to transformation temperature T_g_ for 12 h to release the internal stress from the quench. Next, glasses were cooled down to room temperature and polished to meet the requirements for optical measurements.

X-ray diffraction studies were carried out on the X’Pert Pro X-ray diffractometer supplied by PANalytical (Eindhoven, Netherlands) with Cu K_α1_ radiation (λ = 1.54056 Å) in the 2θ range of 5°–90°. The step size, time per step, and scan speed were 0.017°, 184.79 s, and 0.011°/s, respectively. The X-ray tube was operated at 40 kV and 40 mA and a scintillation detector was used to measure the intensity of the scattered X-rays. In the Appendix A, the schematic of the XRD setup is described in detail. The morphologies of prepared samples were examined using an FEI Company Nova Nano SEM 200 scanning electron microscope (Hillsboro, OR, USA). The MIR spectra of the samples were obtained with a Fourier spectrometer (Bruker Optics-Vertex70V, Rheinstetten, Germany). The measurements were performed using the KBr pellet technique. Absorption spectra were recorded at 128 scans and a resolution of 4 cm^−1^. The MIR spectra were deconvoluted into component bands using the Opus-7.2 program.

The excitation and luminescence spectra of the glasses in the range of 350–750 nm were measured using the Jobin Yvon Fluoromax 4 spectrophotometer (HORIBA, Piscataway, NJ, USA). A PTI QuantaMaster QM40 system (Photon Technology International, Birmingham, NJ, USA) coupled with a tunable pulsed optical parametric oscillator (OPO) pumped by the third harmonic of the Nd:YAG laser (OpotekOpolette 355 LD, OPOTEK,Carlsband, CA, USA) was used for luminescence decay measurements. The laser system was equipped with a double 200 mm monochromator, a multimode UV-VIS PMT (R928), and Hamamatsu H10330B-75 detectors controlled by a computer. Luminescence decay curves were recorded and stored by a PTI ASOC-10 (USB-2500) oscilloscope with an accuracy of ±1 µs.

## 3. Results and Discussion

### 3.1. Structural Studies

The X-ray diffraction method was used to determine the character of the obtained samples. Figure 1 shows the diffractograms of all synthesized samples. It is shown that all diffractograms include an amorphous halo effect ranging between 20° and 30° 2θ, indicating their amorphous character. However, the diffractogram of the GGB20PEu sample (Figure 2), in addition to the amorphous halo, presents reflexes derived from the crystalline phase. The visible reflexes originate from the hexagonal phase of GaPO_4_ [40,41]. The reflexes indicate glass structure ordering along with the addition of P_2_O_5_ and decrease of BaO until the 20 mol% content is reached, and the possible creation of glass–ceramic material. Further replacing BaO with P_2_O_5_ causes disordering of the glass network, which is confirmed by the disappearance of reflexes.

Nonetheless, the observed crystalline phase was not detected under the scanning electron microscope as its crystals were probably too small (Figure 3). Additionally, SEM pictures include artifacts, which are crumbs created during the sample preparation process. The visible bulks are not part of the sample structure, but are unavoidable artifacts created while preparing the sample for testing.

The influence of P_2_O_5_ on the glass structure was analyzed on the basis of deconvoluted MIR spectra of 50GeO_2_-10Ga_2_O_3_-(40−x)BaO-xP_2_O_5_ samples performed in the 400–1700 cm^−1^ range. The MIR spectrum of the glass sample free of phosphorus oxide is presented in Figure 4. The bands between 700 and 1000 cm^−1^ belong to asymmetric stretching vibrations of Ge-O-Ge bonds and symmetric stretching of broken Ge-O^−^ bonds. Bonds between 400 and 700 cm^−1^ can be ascribed to bending vibration of Ge-O-Ge bonds [42]. The MIR spectra of samples free of phosphorus oxide and enriched in 40 mol.% barium oxide show two bands at around 790 and 722 cm^−1^, which are ascribed broken bonds of Ge-O^−^(NBOs) and bonds in [GeO_6_] units, respectively. A less intensive band at around 925 cm^−1^ is associated with asymmetric stretching and Ge-O-Ge bond vibration. The high intensity of the bands corresponding to non-bridging oxygen atom (NBO) vibration resulting in bonds breaking inside the glass network is caused by the high content of barium oxide, which is a glass modifier [43,44,45,46].

The replacement of barium oxide in the amount of 20 mol.% by phosphorus oxide causes the appearance of new bands (1000–1350 cm^−1^ region) related to the bonds in [PO_4_] units and broken P-O^−^ bonds (Figure 5) [47,48]. Additionally, bands at 764 cm^−1^, which derive from Ge-O^−^ bond vibrations and bond vibrations in [GeO_6_] units, are characterized by smaller intensities than the parallel bands from the spectrum in Figure 5. They are caused by the reduction of the content of the glass structure modifier (BaO). Moreover, the band at 1052 cm^−1^ in the spectrum in Figure 5 can be ascribed to NBO vibrations in PO_3_ mode in the Q_1_ tetrahedra unit [49]. These phenomena can be explained by the presence of the high content of barium oxide inside the glass structure and the presence of Ga_2_O_3_ in the glass set, which can be treated as a glass modifier. Additionally, the spectrum in Figure 5 presents sharper bandwidths with a lower value for the full width at half-maximum (FWHM) compared with the other spectra, which could indicate the occurrence of areas of long-range ordering inside the glass structure—the beginning of the crystallization processes.

When the amount of phosphorus oxide exceeds 20 mol.%, the band at around 774 cm^−1^ ascribed to Ge-O^−^ bond vibrations and bond vibrations in the [GeO_6_] units appears again, which could indicate depolymerization processes (Figure 6). Additionally, an increase in the intensity of the band derived from P-O- bond vibration (~1050 cm^−1^) is noticeable (Figure 5). These aspects could be due to the activity of Ga_2_O_3_ as a glass modifier. In general, Ga_2_O_3_ in the glass set can be treated as a glass-forming element and modifier, which depends on its amount and the types of glass elements [50]. The increased molar content of P_2_O_5_ and the reduced content of BaO emphasize the role of Ga_2_O_3_ as a glass modifier, which consequently cause breakage of the bonds in the glass structure. The deconvolution of the spectrum of the sample containing 20 mol.% of phosphorus oxide show the presence of P-O bond vibration in Q_1_ and Q_2_ units at 979 cm^−1^ and 1217 cm^−1^, and 1135 cm^−1^, respectively. The intensity of each band is on similar level, while further addition of phosphorus oxide and complete removal of barium oxide cause dominance of the band intensity derived from Q_2_ units (1178 cm^−1^ and 1301 cm^−1^) over the band intensity derived from Q_1_ units (956 cm^−1^) (Figure 5 and Figure 6). The P_2_O_5_ addition and BaO removal increase the number of connections between [PO_4_] units [47].

Along with the phosphorus oxide addition and the barium oxide removal, the ratio of the intensity of the bands ascribed to P-O and Ge-O bond vibration increases (Figure 4, Figure 5 and Figure 6). The range between 400 and 700 cm^−1^ is the region where bands correspond to the bending of the bridging bond vibrations. Along with phosphorus oxide addition, the intensities of bands between 400 and 630 cm^−1^ decrease parallel with increasing intensities of bands between 630 and 700 cm^−1^, which can be ascribed to the bending vibrations of P-O-P bonds [47,49,51,52].

### 3.2. Photoluminescence Excitation, Emission Spectra, and Luminescence Kinetics

In order to investigate the effect of P_2_O_5_ content on optical properties of GGB glasses doped with europium ions, the excitation spectra monitored at 611 nm were analyzed (Figure 7). It is known that the observed five bands centered at the wavelengths of 362, 382, 393, 415, and 465 nm correspond to transitions from the ground state ^7^F_0_ to the higher energy states ^5^D_4_, ^5^L_7_, ^5^L_6_, ^5^D_3_, and ^5^D_2_, respectively. In the analyzed spectral range of 350–500 nm, two bands are characterized as having the highest intensity. The band at 394 nm (^7^F_0_→^5^L_6_) is most commonly used for optical excitation of glasses doped with Eu^3+^ ions. A second one at 465 nm (^7^F_0_→^5^D_2_), which is dominant in fabricated glasses, is called “hypersensitive transition” and strongly depends on the local environment of the europium ions [22]. In our experiment, the highest intensity of the ^7^F_0_→^5^D_2_ transition was observed in glasses without P_2_O_5_ (GGB0PEu), while the introduction of phosphate oxides led to a slight decrease in excitation intensity up to 30 mol.% for P_2_O_5_. For glasses with maximal content of phosphorous (GGB40P) only, the intensity of the excitation band at 393 nm was on a similar level, but the “hypersensitive transition” intensity was still higher than ^7^F_0_→^5^L_6_ transitions.

Due to the different intensities of ^5^L_6_ and ^5^D_2_ bands, the luminescence spectra in two excitation channels were analyzed. Figure 8a,b presents the normalized luminescence spectra of GGB glasses modified by P_2_O_5_ and doped with 0.2 mol% of Eu_2_O_3_ under 394 and 464 nm laser excitation, respectively. The normalization of all spectra to the ^5^D_0_→^7^F_1_ transition, in which the intensity is independent of the host, provides a useful method for the fast determination of network changes in the vicinity of Eu ions. In both cases, the shape of the luminescence is similar and consists of five emission bands centered at the wavelengths of 575, 589, 611, 650, and 700 nm, originating from ^5^D_0_→^7^F_J_ (J = 0, 1 … 4) transitions. Especially, two of the emission bands shown evident changes resulting from the P_2_O_5_ modification. The first is the ^5^D_0_→^7^F_1_ transition, defined as the magnetic dipole (MD) transition, which is the most intensive transition in the spectra of materials with an inversion symmetry structure [53]. According to the selection rules, the intensity of the ^5^D_0_→^7^F_1_ transition is independent of the ligands of Eu^3+^ ions, hence the spectra normalization is helpful in the characterization of the site asymmetry of europium ions. Second, the ^5^D_0_→^7^F_2_ transition belongs to the electric dipole (ED) transitions and is the dominant emission band in fabricated glasses. The intensity of this transition strongly depends on the local environment of Eu^3+^ ions and is usually called “hypersensitive transition”. In our experiment, the intensity of the emission at 611 nm (^5^D_0_→^7^F_2_) obtained for both excitation routes was changed as a function of the P_2_O_5_ concentration. Based on these changes, the asymmetric ratio (AR) intensities, defined as the relation of the electric dipole transition (^5^D_0_→^7^F_2_) to the magnetic dipole transition (^5^D_0_→^7^F_1_), were estimated (inset in Figure 8). The value of the (^5^D_0_→^7^F_2_)/(^5^D_0_→^7^F_1_) transition ratio gives a factor of the degree of structural distortion from the inversion symmetry of the local environment of the europium ions [39]. We observed that replacing the BaO by P_2_O_5_ up to 20 mol.% leads to a decrease of the asymmetric ratio, which is related to the partial structural arrangement in the ligand field around Eu^3+^ ions. This effect indicates that Eu^3+^ ions could be incorporated into the crystalline phase, which is in good agreement with our structural results (Figure 2 and Figure 5). However, more interesting is the fact that the further increase of phosphate oxide has the opposite effect, whereby the asymmetric ratio increases. This phenomenon confirms that a higher concentration of P_2_O_5_ leads to “depolymerization” of a glassy network, resulting from an increase in the non-bridging oxygen (NBO) group [37]. In this way, the phosphorous oxide can change its role from being a glass modifier to a glass-forming compound.

The luminescence decay curves of GGBxPEu glasses doped with Eu^3+^ ions obtained under two excitation wavelengths (λ_exc_ = 394 and 464 nm) in a semi-logarithmic scale are presented in Figure 9. All obtained results are described by single-exponential approximation. The shortest lifetime of the ^5^D_0_ state of Eu^3+^ was obtained for the glass without phosphorous oxide, which was similar to the other germanate-based glasses from the literature [11,23]. It is worth noting that the partial replacement of BaO by P_2_O_5_ in the glass network caused an almost two-fold increase in the luminescence lifetime of the ^5^D_0_ state of Eu^3+^, increasing from 1.32 ms (GGB0PEu) to 2.19 ms (GGB20PEu). However, when the BaO was completely substituted by P_2_O_5_, the luminescence lifetime decreased slightly (Table 1). Additionally, the measured values of the luminescence lifetimes of the ^5^D_0_ state for glasses with phosphorous oxide are comparable to the lifetimes in other phosphate glasses [27].

Due to the similar changing trends for the luminescence lifetime for both excitation wavelengths, we concluded that fabricated glasses are characterized by good homogeneity of Eu^3+^ centers (insets in Figure 9).

Further detailed analysis of the luminescence shape showed interesting effects in GGB glasses caused by partial replacement of BaO by P_2_O_5_ (Figure 10). Here, to compare the luminescence shape changes more easily, all spectra at the ^5^D_0_→ ^7^F_2_ transition were normalized. To correlate structural changes with the luminescent properties of investigated glasses, we focused precisely on the changes of two transitions: (i) ^5^D_0_→ ^7^F_2_, where the incorporation of P_2_O_5_ leads to a decrease of the spectral line bandwidth from 15 nm to 10 nm and a 2 nm shift of the emission peak towards shorter wavelengths; (ii) ^5^D_0_→^7^F_1_, where the relation between the Stark sublevel presents a considerable number of slightly different sites for Eu^3+^ centers [54]. In the case of spectral broadening of the emission band at the wavelength of 611 nm (^5^D_0_→^7^F_2_), the narrowing effect is connected with the higher heterogeneity of glassy structure in samples with a higher concentration of P_2_O_5_ [55,56]. Additionally, the high phonon energy of P-O bonds minimizes the energy migration between europium ions, hence the inhomogeneous broadening is lower. The blue-shift in the peak emission wavelength can be ascribed to the formation of non-bridging oxygen groups more than bridging oxygen groups in the GGBxP glass sample.

According to crystal field theory, the next analyzed ^5^D_0_→^7^F_1_ transition directly shows the splitting of the ^7^F_1_ level. If the ^7^F_1_ level is not split then cubic or icosahedral crystal fields are observed. For hexagonal, tetragonal, and trigonal crystal fields, the ^7^F_1_ level is split for two components. In the case of the lowest symmetries (orthorhombic crystal fields), three sublevels occur in the ^7^F_1_ level [53,57]. The inset in Figure 10 shows a deconvoluted ^5^D_0_→^7^F_1_ transition for GGB0P and GGB40P glasses. In our case, germanate glass without P_2_O_5_ is characterized by low symmetry and three Stark sublevels bands are easy to determine. Additionally, the intensities of all sub-bands are approximately the same. It is worth noting that replacing BaO with 40 mol% of P_2_O_5_ leads to visible domination of the central emission sub-band. The obtained results confirm that europium ions in fabricated germanate glass can occupy two or three sites of symmetry, depending on the P_2_O_5_ concentration.

## 4. Conclusions

The germanate glasses with 50GeO_2_-10Ga_2_O_3_-(40−x)BaO-xP_2_O_5_-0.2Eu_2_O_3_ molar compositions have been synthesized and investigated by XRD, SEM, MIR, and optical spectroscopy. MIR spectra analysis has shown that phosphorous oxide (up to 40 mol.%) introduces additional bond vibration as ion co-forming in the network of a fabricated glass. The analysis of intensity changes of absorption bands confirmed that the reduction of barium oxide content was evident with a decrease in the intensity of bands related to Ge-O- vibrations and the disappearance of the band ascribed to P-O- bond vibration. The luminescence studies carried out in two excitation routes for Eu^3+^ ions at 394 nm (^7^F_0_→^5^L_6_) and 464 nm (^7^F_0_→^5^D_2_) showed that replacing BaO with up to 20 mol.% of P_2_O_5_ leads to a decrease in the luminescence asymmetric ratio, which is in good agreement with structural results (XRD, MIR). In general, the values for asymmetric ratio suggest that in all samples the Eu^3+^ ions are located at sites without inversion symmetry. However, partial local long-range ordering was observed in GGB20PEu glass. Based on detailed analysis of the luminescence shapes of two main radiative transitions of Eu^3+^ ions, the spectral narrowing of the band at 611 nm (^5^D_0_→^7^F_2_) and evident splitting of the emission band at 589 nm (^5^D_0_→^7^F_1_) confirm that fabricated glasses are characterized by low symmetry (orthorhombic) of the local environment of europium ions. The obtained results indicate that chemical modification of germanate glass doped with europium ions by phosphorous oxide allows the design of functionalized luminescent properties, which are important for applications in modern light-emitting devices. In future work, we will extend our experiments towards glass–ceramic materials (especially in glass samples with visible nanocrystalization, such as GGB20PEu) and we will show the possibility of tuning the emissions from Eu^3+^ ions resulting from the nanostructurization of the local network, as well as the impact of the activator content.

## Figures and Tables

**Figure 1 materials-13-02817-f001:**
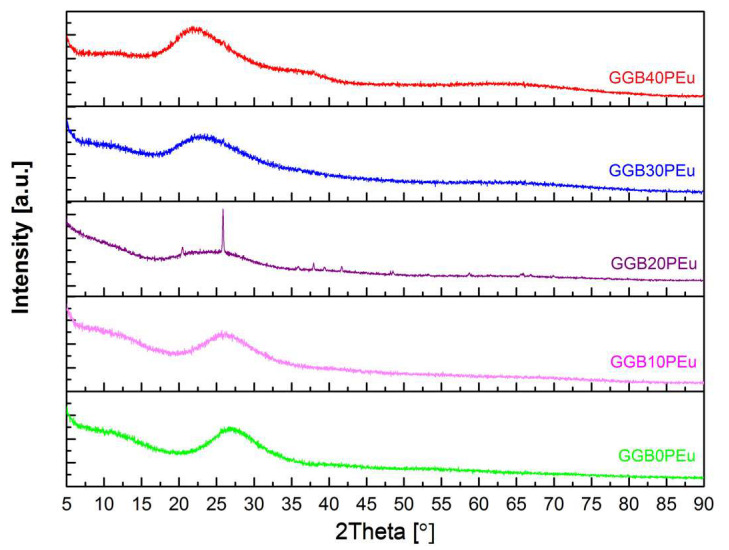
Diffractograms of the GGBxP_Eu glasses.

**Figure 2 materials-13-02817-f002:**
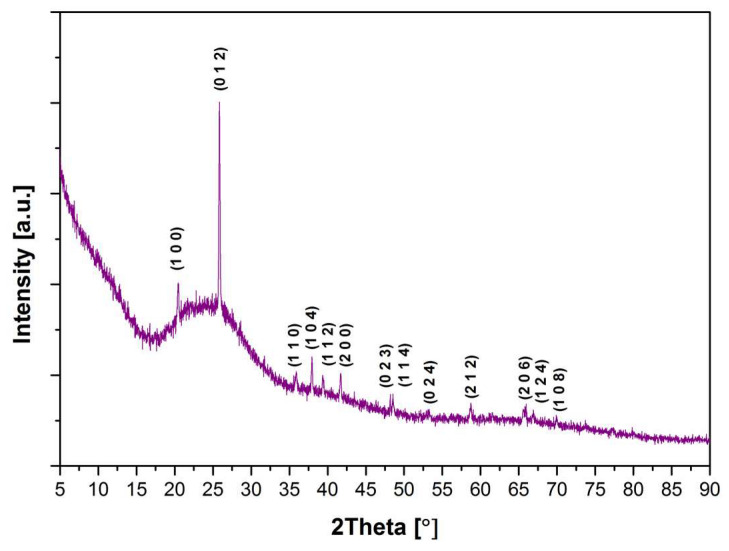
XRD pattern of the GGB20PEu glass sample with hkl crystalline planes for GaPO_4_.

**Figure 3 materials-13-02817-f003:**
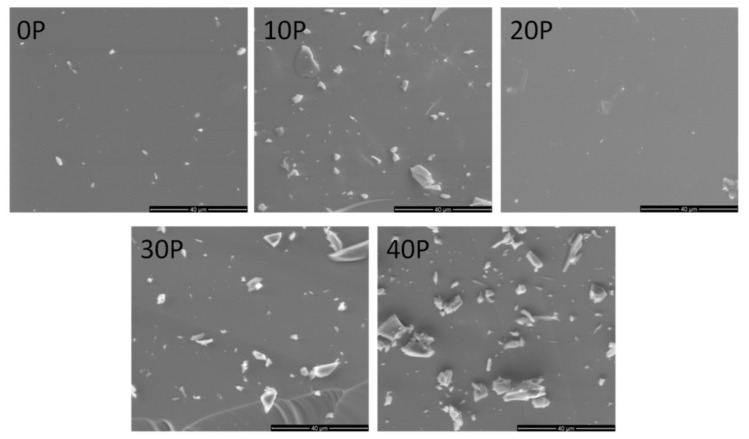
Scanning electron microscope images of samples from the GGBxP_Eu system (xP determines the O_5_ in the sample)

**Figure 4 materials-13-02817-f004:**
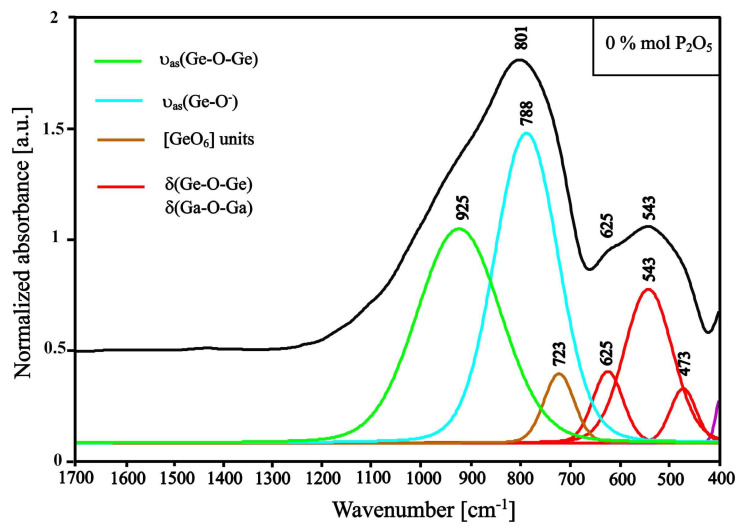
Deconvoluted spectrum of GGB0PEu glass sample (without P_2_O_5_).

**Figure 5 materials-13-02817-f005:**
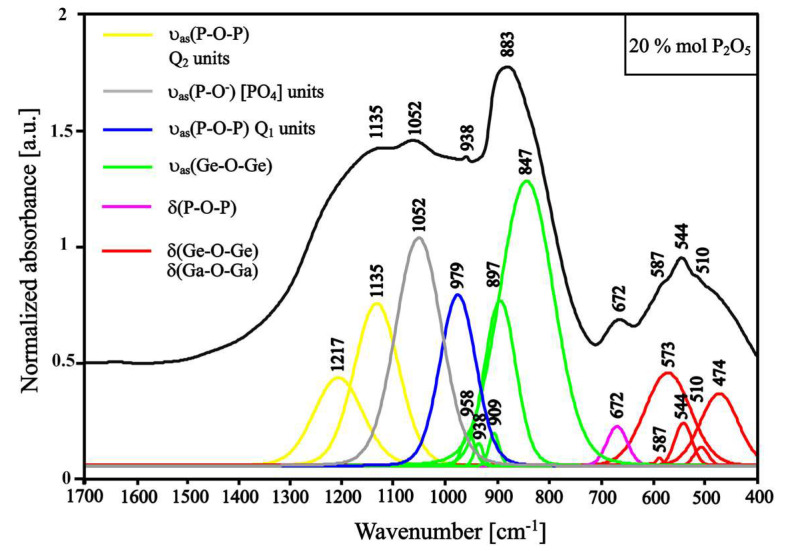
Deconvoluted spectrum of GGB20PEu glass sample (with 20 mol.% of P_2_O_5_).

**Figure 6 materials-13-02817-f006:**
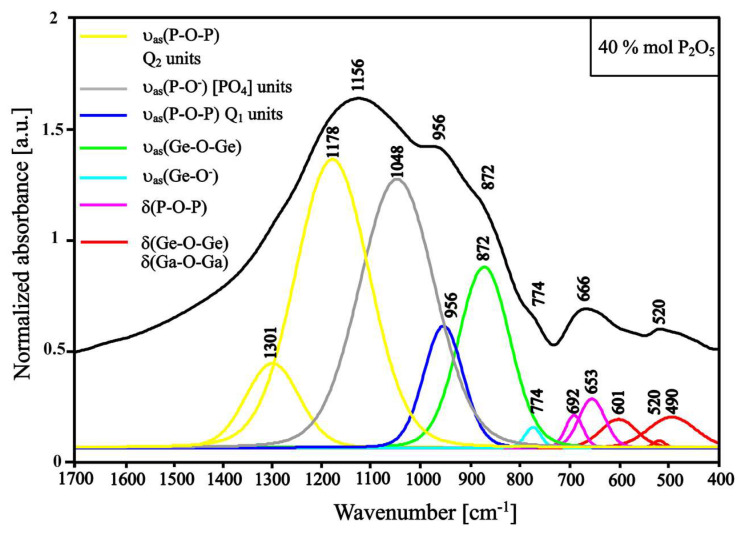
Deconvoluted spectrum of GGB40PEu glass sample (with 40 mol.% of P_2_O_5_).

**Figure 7 materials-13-02817-f007:**
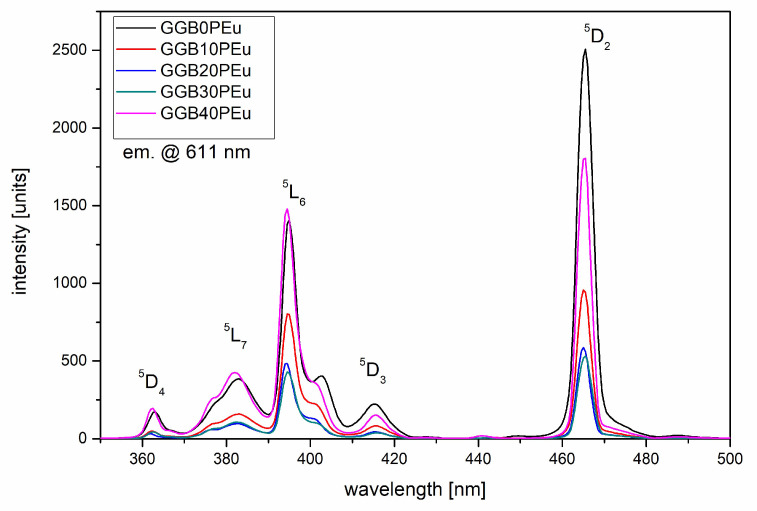
Excitation spectra (monitoring at 611 nm) of Eu^3+^ ions doped GGBxPEu glass modified by P_2_O_5_.

**Figure 8 materials-13-02817-f008:**
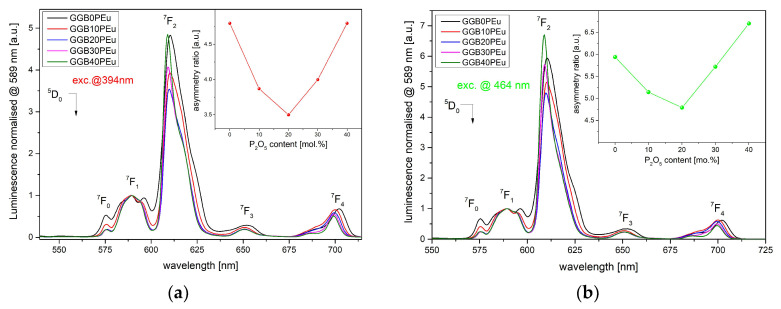
Luminescence spectra of Eu^3+^-ion-doped GGBxPEu glasses (**a**) under 394 and (**b**) under 465 nm of laser excitation. (Inset in both figures) Asymmetry ratio of Eu-doped GGBxPEu glass modified by P_2_O_5_.

**Figure 9 materials-13-02817-f009:**
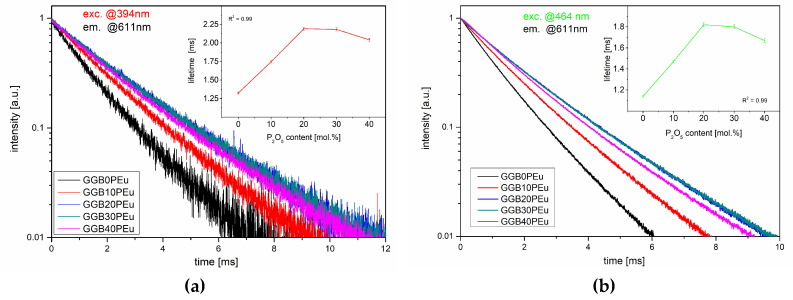
Luminescence decays of the ^5^D_0_ energy level of europium ions in fabricated GGBxPEu glasses (**a**) under 394 nm laser excitation and (**b**) under 464 nm of laser excitation. (Inset) The changes in the lifetime as a function of P_2_O_5_ concentration.

**Figure 10 materials-13-02817-f010:**
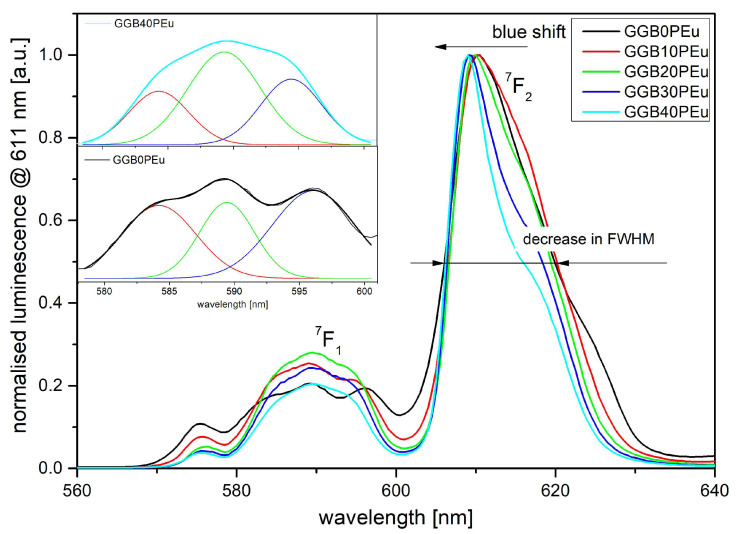
Normalized luminescence spectra of Eu^3+^-doped GGBxP glasses under 394 nm laser excitation. (Inset) Deconvolution of ^7^F_1_ multiplet for glasses with 40 mol% of P_2_O_5_ (up) and without P_2_O_5_ (down). FWHM, full width at half-maximum.

**Table 1 materials-13-02817-t001:** The ED/MD transition ratio and the τ_394nm_ and τ_464nm_ lifetime values of the ^5^D_0_ level of Eu^3+^ ions.

Glass Sample	ED/MD Ratio		τ_394nm_ [ms] ± 0.01 ms	τ_464nm_ [ms] ± 0.01 ms
	exc. @394 nm	exc.@464 nm		
GGB10P	4.8	5.9	1.32	1.14
GGB10P	3.9	5.1	1.74	1.47
GGB20P	3.5	4.8	2.19	1.82
GGB30P	4	5.7	2.18	1.80
GGB40P	4.8	6.7	2.04	1.67

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
