# Peer review of "Luminescent Studies on Germanate Glasses Doped with Europium Ions for Photonic Applications"

_materials, 2020, doi:10.3390/ma13122817_

Round 1
Reviewer 1 Report
The manuscript proposed the preparation of Germanate glasses based phosphers with detailed characterization and useful discussion. However, there are some minor issues need to be corrected before the acceptance of the paper. Because of these, the manuscript cannot be accepted in its present form. I suggest that the authors should response the following queries and comments, which are listed below.
(1) (line 74)
The word of MIR need to be defined in the first time.
(2) (Figure 1)
Please index the hkl crystalline planes for GaPO4, and define 0P, 10,P and so on.
(3) (Figure 2)
More description is required for the SEM. For examples, what are particles and bulks in the Figure 2?
(4) (Figure 8)
The unit of intensity needs to change from “a. u.” to “unit”
(5) (Table 1)
It seems that ED/MD and lifetime are not in a direct relationship. Please explain.
Author Response
Dear Reviewer,
Thank you for your valuable remarks. In attached file we put our answers.

Reviewer 2 Report
The work from Zmojda et al represents a good improvement in the studies of gain materials for photonic systems. The authors have shown local asymmetry ratio confirming higher changes in local symmetry around Eu3+ ions. Thy have also shown that germanate glass doped with europium ions with functionalized luminescent properties are a promising material for certain wavelengths.
I would suggest to the authors mentioning the market of Eu3+ ions.
Add a Schematics of the set-up in Figure 1 for the diffractograms.
Extend the explanation after line 158 of figure 6.
One minor spell error in the last sentence, correct by: Intense.
Reviewer 3 Report
The authors describe an article entitled " Luminescent Studies on Germanate Glasses doped with Europium Ions for Photonic Applications". The topic of the manuscript is interesting, and the manuscript constitutes an interesting study concerning the development of luminescent materials from Rare Earth (RE) ions.
The work is well written, and sufficient spectra and figures are included in the manuscript for comprehension and clarity. Convincing results have been obtained in this work. Overall, I think that this is a good manuscript, that I recommend for publication after inclusion of minor revisions.
1) What about the nanostructuration of the glasses surfaces. AFM measurements should be done;
2) What about the electrochemical properties of the glasses ?
For all the above-mentioned reasons, at present, do not publish
Reviewer 4 Report
The manuscript describes the luminescent studies on germanate glasses doped with Europium ions for photonic applications.
The paper needs improvements before publication. It is not easily to read and is repetitive in experiments and presentation of these. Five figures describes similar findings without systematization of these results.
The introduction do not present well the progress beyond the state of the art introduced by this paper. Please improve.
The spectra presented in the Figure 3 are normalized? The Absorbance scale must be included. It is necessary for better understanding of the phenomena.
The legend of the Figure 3 is not clear and it must be included. Is not clear to what is corresponding each spectrum.
The values of the peaks of the black curve must be also included.
Exactly the same remarks as the previous one for the Figures 4, 5, 6 and 7.
The text and the Figures are too repetitive. Maybe presenting the first and the last figure will be enough. I recommend this reduction of the numbers of the Figures and a schematic evolution of the peaks in function of composition will be more useful to fully understand the modifications of the properties.
In figure 10 the errors bars are necessary to be included. Furthermore, a statistical method must be applied in order to prove that the data obtained are significantly different from statistical point of view.
In all figures where the spectra were normalized what method was used? Please include some comments in the paper. Why normalization of the spectra was necessary.
No applications of these materials are included in the paper. It is necessary.
The conclusions are not clear enough regarding the findings of this research work. The future possible application should be presented.
Reviewer 5 Report
In this work, the authors investigated the role of P2O5 on luminescent and structural properties of GGB glasses doped with Eu3+ ions. I think this manuscript provides a systematic introduction of P2O5 using in GGBxPEu glasses. I recommend the publication of this work on Materials after addressing the following issues:
(1) In Figure 1, the authors depicted diffractograms of the GGBxP_Eu glasses. However, some diffraction peaks located at 15-50° cannot be distinguished deeply. I think there should have enlarged view of the diffraction peaks of 0P-40P in order to recognize some specific differences between 0P-40P.
(2) The colors of different spectrums in Figure 3 to 7 are not easy to identify. I think the colors need to be adjusted to obtain more artistic deconvoluted spectrums.
(3) In the conclusion part of this manuscript, the authors demonstrated “The obtained results indicate that germanate glass doped with europium ions with functionalized luminescent properties is a potential candidate for application in modern photonic devices.” (Page 10, Line 285) Nevertheless, the manuscript has no comparative tests about the germanate glasses with and without Eu3+. I think the controlled trial should be carried out to emphasize the importance role of Eu3+.
Reviewer 6 Report
Jacek Żmojda reported work "Luminescent Studies on Germanate Glasses doped with Europium Ions for Photonic Applications" can be acceptable with corrections as mentioned below.
1) I guess in this authors did not demonstrate the applications, hence title of the manuscript should be modified accordingly.
2) Eu3+ doping and their amount (quantifications on glass surface) can be more precisely validated through XPS spectra. I recommend authors to perform XPS studies with doped and undoped glass.
3) By looking at SEM images crystallinity of glass at the surface were in randomized, please provide with EDS analysis graph to validate it. Why authors could not see any signals from 20P SEM images in Figure 2
Note: It is very hard to follow up authors notation in images. Can authors telling 0P, 10P and 20P what are these notations ?? can authors specify it ?
4) It is better to provide systematic reduction of BaO amount in glass and its effects rather than direct demonstrations with specified amount and place it accordingly. Because of some characterisation studies in the initial stage and some of them are in final stage (Specially BaO dependent). Typically Ba2+ ions can affect the crystallinity in glass. Did authors reported those effects previously If yes please cite it wherever it is appropriate.
5) It is better to provide all luminescence spectra with specific fixed range. Why authors presented Fig. 11 from 560 to 640, but others are different.
Round 2
Reviewer 4 Report
The manuscript was improved by revision, fhe answeers are enough good and the paper could be published without further modifications.